**Data Availability Statement:** Ontario Health is prohibited from making the data used in this research publicly accessible if it includes potentially

# Scheduled and urgent inguinal hernia repair in Ontario, Canada between 2010 and 2022: Population-based cross sectional analysis of trends and outcomes

**Steven Habbous**[1,2]*, **David Gomez**[3,4], **David Urbach**[3,5], **Erik Hellsten**[1]

**1** Ontario Health, Toronto, Ontario, Canada, **2** Epidemiology & Biostatistics, Western University, London, Ontario, Canada, **3** Department of Surgery, Temerty Faculty of Medicine, University of Toronto, Toronto, Ontario, Canada, **4** Division of General Surgery, St Michaels Hospital, Unity Health, Toronto, Ontario, Canada, **5** Women's College Hospital, Toronto, Ontario, Canada

\* Steven.habbous@ontariohealth.ca

## Abstract

### Introduction

We examine trends in inguinal hernia repairs with respect to the COVID-19 pandemic and secular trends in Ontario, Canada.

### Methods

This was a retrospective cohort study. Hernia repairs performed January 1, 2010-December 31, 2022 were captured from health administrative inpatient and outpatient databases. Patients managed in three clinical settings were examined: public hospital in-patient, semi-private hospital in-patient (Shouldice Hospital), and public hospital out-patient. We examined the effect of the COVID-19 pandemic on surgical volumes, clinical setting, patient characteristics by setting, time from diagnosis until surgery, hospital length-of-stay, and patient outcomes (90-day readmissions, 1-year reoperations). We used multivariable logistic regression to examine whether patient outcomes were comparable between the COVID-19 period and the pre-pandemic period, adjusted sociodemographic and clinical factors. Shouldice Hospital is the only semi-private hospital in Ontario specializing in hernia repair (patients pay for the mandated admission, but not for the procedure).

### Results

During the pandemic (March 2020-December 2022), there were 8,162 fewer (15%) scheduled inguinal hernia repairs than expected, but the age-sex standardized rate of urgent repairs remained unchanged. Shouldice Hospital performed more surgeries in the COVID-19 era than pre-pandemic and had a shorter average LOS by 24 hours, despite treating more patients with older age, higher ASA score [adjusted odds ratio (aOR) 2.13 (1.93–2.35) III vs I-II] and greater comorbidity [aOR 1.36 (1.08–1.70) for 2 vs none] than pre-pandemic. Patients treated in the COVID-19 era experienced a longer time until surgery, being the

identifiable personal health information and/or personal information as defined in Ontario law, specifically the Personal Health Information Protection Act (PHIPA) and the Freedom of Information and Protection of Privacy Act (FIPPA). Due to these legal and ethical restrictions, data will not be made publicly available. However, upon request, data de-identified to a level suitable for public release may be provided [contact: Datarequest@ontariohealth.ca].

**Funding:** The author(s) received no specific funding for this work.

**Competing interests:** The authors have declared that no competing interests exist.

longest in 2022 (median 133 days). Ninety-day readmissions and 1-year reoperations were lower in the COVID-19 era and lower for patients receiving surgery at Shouldice Hospital.

## Conclusion

During the COVID-19 pandemic, there were 8,162 fewer scheduled hernia repairs than expected, longer wait-times until surgery, shorter length-of-stay, and more patients with comorbidities, but outcomes were not worse compared with the pre-pandemic period.

## Introduction

During the COVID-19 pandemic, many scheduled surgeries were deferred to increase capacity for patients with SARS-CoV-2 infection requiring in-hospital care [1, 2]. After government-mandated restrictions on surgical activity were lifted, ongoing surges in SARS-CoV-2 infection and hospital staffing challenges resulted in continued reductions in surgical activity. As a result, the rate ratio of scheduled inguinal hernia repair during the first year of the pandemic decreased by 21% without a concomitant rise in emergency department (ED) visits for urgent hernia repair, need for hospitalization, length of stay, intervention modality, or mortality [3]. While these results are reassuring, patients waiting for scheduled hernia repair may exhibit worse quality of life due to persistent or worsening symptoms or changes in hernia characteristics.

One solution to catch up on the surgical wait-list while safeguarding inpatient capacity is to transition some inpatient procedures to the outpatient setting whenever possible. Hernia repair is one candidate for this transition and can safely be performed in an outpatient setting for most patients [4]. There has already been a general shift towards outpatient primary inguinal hernia repair in several countries as well as increased use of laparoscopy [4, 5]. Another approach to reducing surgical waitlists is the increased utilization of out-of-hospital facilities or surgical centres, which allow for surgeries to be performed outside of general hospital settings and their attendant COVID-19-related pressures. In Ontario, the Shouldice Hospital is the only surgical centre specializing in hernia repairs performed on an exclusively inpatient basis [6]. As a long-standing specialty centre, the Shouldice Hospital is unique and performs a large percentage of all inguinal repairs in the province [6]. While the procedure itself is publicly funded through the Ontario Health Insurance Plan (OHIP), patients are required to pay out-of-pocket for the subsequent inpatient stay.

In the present study, we examined the patterns and outcomes of inguinal hernia repair in Ontario over a 12-year span and whether these have been impacted by the COVID-19 pandemic. We examine where patients receive surgery, contrasting those treated at public hospitals (inpatient or outpatient) and those treated at the Shouldice Hospital. We assessed the change in patient characteristics and outcomes over time to ensure that the quality of care was not adversely affected due to hospital pressures imposed by the COVID-19 pandemic.

## Methods

### Patient selection

Inguinal hernia repairs were identified from the Canadian Institute for Health Information Discharge Abstract Database (CIHI-DAD) (inpatient procedures) and National Ambulatory Care Reporting System (CIHI-NACRS) (outpatient procedures), where the primary intervention coded was hernia repair (Canadian Classification of Interventions code 1SY80), the

intervention was not abandoned (status attribute not 'A'), the surgery was performed in-hospital (out-of-hospital indicator not 1), the most responsible diagnosis was an International Classification of Diseases (ICD, 10th revision) for inguinal (K40), and the location attribute was not 0 (not hernia repair) (Fig S1 in S1 File). We examined procedures performed between January 1, 2010 and December 31, 2022 and restricted to adults (age ≥18). Data were extracted May 2, 2023.

## Setting

The repair setting was classified as outpatient if captured by the CIHI-NACRS database and inpatient if captured by CIHI-DAD. Inpatient procedures were further subclassified into 3 distinct groups: 1) Shouldice Hospital if the procedure was performed there (all non-urgent); 2) non-urgent inpatient repairs performed at a public hospital; 3) and urgent inpatient repairs (all performed at a public hospital). Urgent operations were defined as an admission with entry code = E (admitted through the emergency department), admission category = U (urgent), or if the patient arrived by ambulance.

## Covariates

Repairs were classified as laparoscopic vs. open and mesh vs. no mesh (Table S1 in S1 File). Comorbidity was characterized with the Charlson Comorbidity Index, using ICD-10 codes for hospital encounters occurring within 3 years prior to the hernia repair. ICD-10 codes indicating diabetes were supplemented with prescriptions for anti-diabetic agents using the Ontario Drug Benefits program for patients age 65+ years. Patients were classified as having obesity if the anesthesiologist or physician billed OHIP for E010 (body mass index >40) or E676 (morbid obesity) anytime within 1 month prior to the hernia repair date. Patients were similarly classified according to the American Society of Anesthesiologists (ASA) scoring system as ASA III-V if the anesthesiologist or physician billed for E022 (ASA III), E017 (ASA IV), or E016 (ASA V), and ASA I-II otherwise [7]. Patient sociodemographics (age, sex) were obtained from the Registered Persons Database (RPDB). The postal code from the RPDB was linked to the Ontario Marginalization Index via the Postal Code Conversion File (PCCF+) to obtain neighbourhood-level sociodemographic characteristics [8]. Travel distance was computed as the Euclidean distance using the latitude and longitude of the patients' residence and the hospital where the surgery was performed using the PCCF+.

## Outcomes

As a surrogate measure of timely access to surgery, we estimated the interval between decision to operate and surgery date. We searched the OHIP physician billing database for the diagnostic code 550–553 (ICD-9 hernia diagnosis code) within 1-year prior to the surgery date. The earliest service date with this diagnostic code defined the decision to operate date.

As a short-term outcome, the readmission rate was estimated using the DAD for an admission (any cause) within either 90 days following the hernia repair (for outpatient repairs) or 90 days following hospital discharge (for inpatient repairs). As a longer-term outcome, we measured the rate of re-operations (e.g. due to recurrence or treatment failure) within 1 year after repair (restricted to repairs occurring no later than December 31, 2021 to ensure sufficient follow-up). For this outcome, we restricted our cohort to first-ever hernia repairs by excluding any repair associated with the OHIP physician billing add-on code E725 (recurrent–all types of hernia) or E726 (repeat recurrent inguinal hernia). Since laterality was unavailable, re-operations for hernia repair were considered if they occurred >7 days after the primary repair (to account for potential delays in repair for bilateral hernias).

## Statistical methods

Temporal trends were presented using descriptive statistics. For estimates of population rates of hernia repairs, we computed age-sex-standardized rates using population counts over time and reported surgical rates per 10,000 population, directly standardized to the 2019 population (age-strata were 18–29, 30–44, 45–59, and 60+. To avoid zero-strata counts, 18–44 years was used for urgent inguinal repairs (females only) [9].

To compare the volume of hernia repairs performed during the pandemic period with the expected volume based on pre-pandemic trends, we regressed the weekly surgical count on year, month, and week number (1–52) for repairs between 2014 and 2019. This model was extrapolated until December 31, 2022. The difference between the observed and expected number of weekly surgeries was computed and regressed on time period, categorized as wave 0 (pre-pandemic), wave 1 (February 26, 2020 –August 31, 2020), wave 2 (Sept 1, 2020 –Feb 28, 2021), wave 3 (Mar 1, 2021 –July 31, 2021), wave 4 (Aug 1, 2021 –December 14, 2021), wave 5 (December 15, 2021 –March 14, 2022), and wave 6 (March 15, 2022 –December 31, 2022) [10]. The coefficients and standard errors for the intercept term (wave 0) and each level of the time period were multiplied by the number of weeks corresponding to the time period (weekly counts become time period counts). The coefficients represent observed minus predicted and represents a less-than-expected (negative) or greater-than-expected (positive) number of repairs during the time period [2]. To compute the overall number greater or less than expected during the pandemic, the time-period coefficients were summed and the time-period standard errors (SE) combined as $95\% \; CI \; = \; estimate \pm \sqrt{\sum \left( \frac{uCL-lCL}{2} \right)^2}$.

Factors associated with 90-day readmissions and 1-year reoperations were assessed using logistic regression, reporting adjusted odds ratios (aOR) with 95% confidence intervals (CI). Unless otherwise stated, all models were adjusted for age, sex, obesity indicator, ASA score, Charlson comorbidity score, hospital setting, repeat repair, rurality, and neighbourhood-level deprivation and ethnic diversity. Models stratified by setting were also adjusted for use of mesh and laparoscopic approach. Analyses were performed on complete case subsets.

## Privacy

All analyses were performed at Ontario Health using SAS version 9.4 (SAS Institute Inc., Cary, NC). P-values <0.05 were considered statistically significant, but interpreted cautiously owing to the large sample size. This study was compliant with section 45(1) of PHIPA (Ontario Health is a prescribed entity): ethics review was not required as per the privacy assessment at Ontario Health. Data were analyzed using the Analytics Data Hub at Ontario Health with patient identifiers removed or pseudonymized prior to access.

## Results

A total 257,517 inguinal hernia repairs (239,858 unique patients) were captured between January 1, 2010 and December 31, 2022 and included in the study (Fig S1 in S1 File). Patients were a mean 59.9 (SD 15.6) years of age at the time of repair, were predominantly male (92%), and treated in an outpatient public-hospital setting (71%) (Table 1).

### Surgical volume and rate

The annual volume of scheduled inguinal hernia repairs remained steady between 2010 (n = 19,714) and 2019 (n = 19,402), but the number of urgent repairs increased over time (Fig 1A). After age-sex standardization to account for changing demographics, the rate of scheduled inguinal hernia repairs decreased from 19.7 (SE 0.14) per 10,000 in 2010 to 16.2 (SE 0.12)

**Table 1. Cohort characteristics.**

|  | N (%) or mean (SD)<br>N = 257,517 |
|---|---|
| Age (years) at time of repair | Mean 59.9 (SD 15.6) |
| Setting |  |
| Outpatient (public hospital; scheduled) | 183,990 (71%) |
| Inpatient (Shouldice Hospital; scheduled) | 57,755 (22%) |
| Inpatient (public hospital; scheduled) | 7,797 (3%) |
| Inpatient (public hospital; urgent) | 7,975 (3%) |
| Sex |  |
| Male | 238,077 (92%) |
| Female | 19,440 (8%) |
| American Society of Anesthesiologists score |  |
| I-II | 170,157 (66%) |
| III | 73,287 (28%) |
| IV | 13,971 (5%) |
| V | 102 (<1%) |
| Obesity |  |
| No | 255,135 (99%) |
| Yes | 2,382 (1%) |
| Comorbidity score |  |
| 0 | 211,996 (82%) |
| 1 | 33,177 (13%) |
| 2 | 7,591 (3%) |
| 3+ | 4,753 (2%) |
| Travel distance (km) | Mean 56.9 (SD 166.1)<br>Median 13.5 (IQR 4.9, 39.0) |
| Type of repair |  |
| Open (other technique) | 159,484 (62%) |
| Open (special technique) | 59,986 (23%) |
| Laparoscopic | 38,047 (15%) |
| Use of mesh |  |
| Mesh | 182,620 (71%) |
| No mesh | 74,897 (29%) |
| Repair order |  |
| Primary repair | 237,299 (92%) |
| Recurrent repair | 20,218 (8%) |
| Rurality |  |
| Urban | 219,801 (86%) |
| Rural | 36,077 (14%) |
| Missing | 1,639 |
| Deprivation |  |
| 1 (least marginalized) | 60,970 (24%) |
| 2 | 54,719 (22%) |
| 3 | 49,537 (20%) |
| 4 | 46,362 (18%) |
| 5 (most marginalized) | 41,508 (16%) |
| Missing | 4,421 |
| Ethnic diversity |  |
| 1 (least diverse) | 56,363 (22%) |

(*Continued*)

**Table 1.** (Continued)

| | N (%) or mean (SD) N = 257,517 |
|---|---|
| 2 | 51,502 (20%) |
| 3 | 49,434 (20%) |
| 4 | 48,431 (19%) |
| 5 (most diverse) | 47,366 (19%) |
| Missing | 4,421 |

Column percents are shown, except where indicated

SD–standard deviation; IQR–interquartile range

in 2019 without an associated increase in the rate of urgent inguinal hernia repairs over the same time period [0.54 (SE 0.02) per 10,000 in 2010 and 0.55 (SE 0.02) per 10,000 in 2019) (Fig 1B).

At the onset of the pandemic (March 2020), as expected due to government-mandated slowdowns in scheduled surgery, the number and rate of hernia repairs decreased for scheduled repairs, without an associated increase in urgent repairs (Fig 1). Compared with the expected numbers of scheduled hernia repairs during the pandemic (n = 54,869), there were 46,702 actual surgeries performed (8,167 fewer than expected; a 15% reduction). This was statistically significant: estimate 8,162 (95% CI 6,411 to 9,912) fewer scheduled inguinal repairs (Table 2; Fig 2). There appeared to be some temporary "catch-up" during wave 2 (483 more repairs than expected), but this did not reach statistical significance and therefore not more than expected (95% CI -249 to 1,215).

In 2020, the number of scheduled repairs decreased in all settings (Fig S2 in S1 File), corresponding to a decline in outpatient procedures in 2021 that was partially offset by a rise in inpatient procedures performed at the Shouldice Hospital. Shouldice Hospital partly increased their procedural output by performing more operations on Fridays during the pandemic (1% in 2019; 5% in 2020; 9% in 2021; 8% in 2022).

## Hospital setting and patient characteristics

The proportion of scheduled hernia repairs performed in an outpatient setting increased between 2010 (69%) and 2019 (77%) (Fig 3). Of note, 1 in 4 scheduled inguinal hernia repairs were performed at Shouldice Hospital, which follows a long-established model of care mandating an inpatient hospital stay for every case, driving down the overall provincial rate of outpatient scheduled hernias. Over the entire study period, 96% of all scheduled inguinal hernia repairs performed at a public hospital were performed as outpatient cases, despite some regional variability (Fig 4).

The characteristics of patients receiving care differed by setting (Table 3). Compared with scheduled outpatient repairs, inguinal hernia patients receiving treatment at Shouldice Hospital were younger (mean 57.1 versus 59.9 years), less likely to be female (4.5% versus 8.0%), had an ASA score I-II (94% versus 61%), more likely to have no comorbidity (87% versus 83%), did not have obesity (0% versus 1.1% had obesity), were less often recurrent repairs (5.2% versus 12%), and were more likely to reside in the least deprived quintile of neighbourhoods (31% versus 22%). Conversely, scheduled patients who required admission to a public hospital were older (70.6 years), had higher ASA score (24% with ASA I-II vs. 61%), greater comorbidity (55% had no comorbidity vs. 83%), and 2.6% of patients had obesity. The median travel distance to receive surgery was similar between settings at public hospitals (median 9.3km for

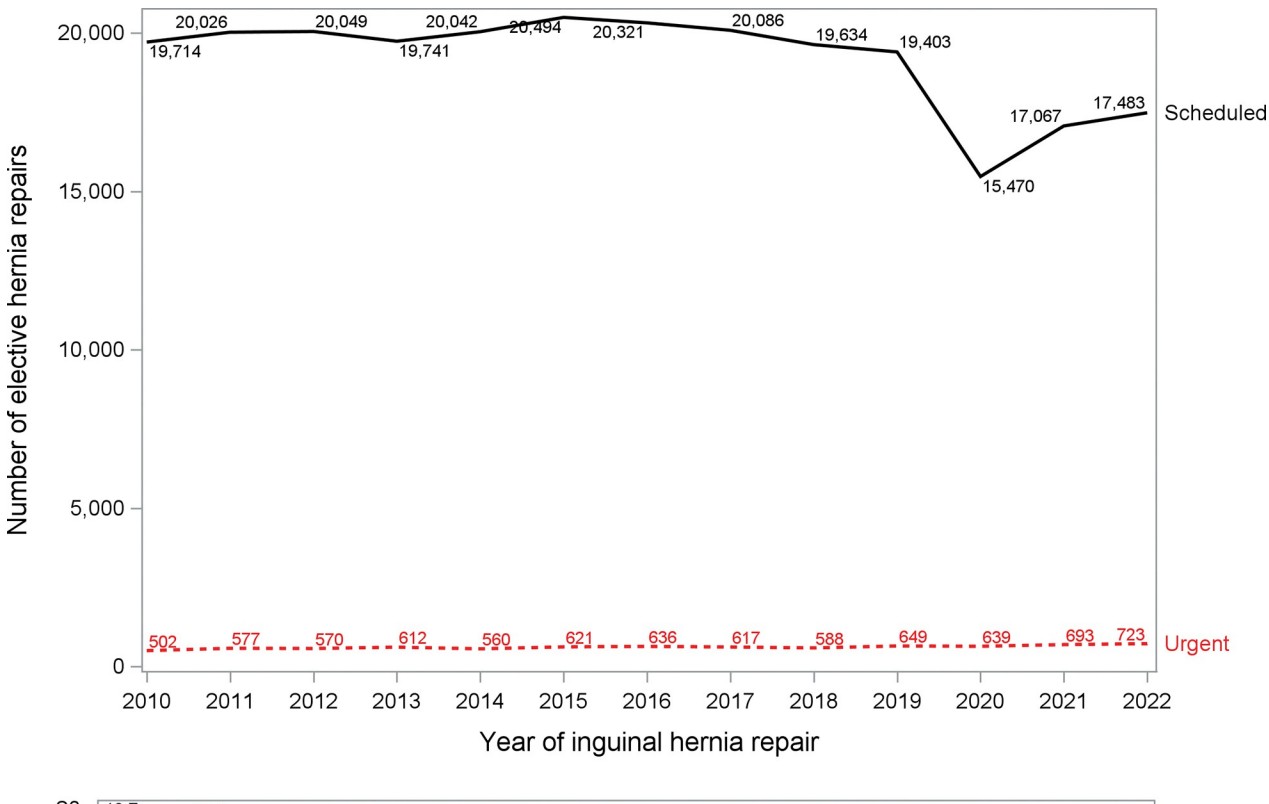

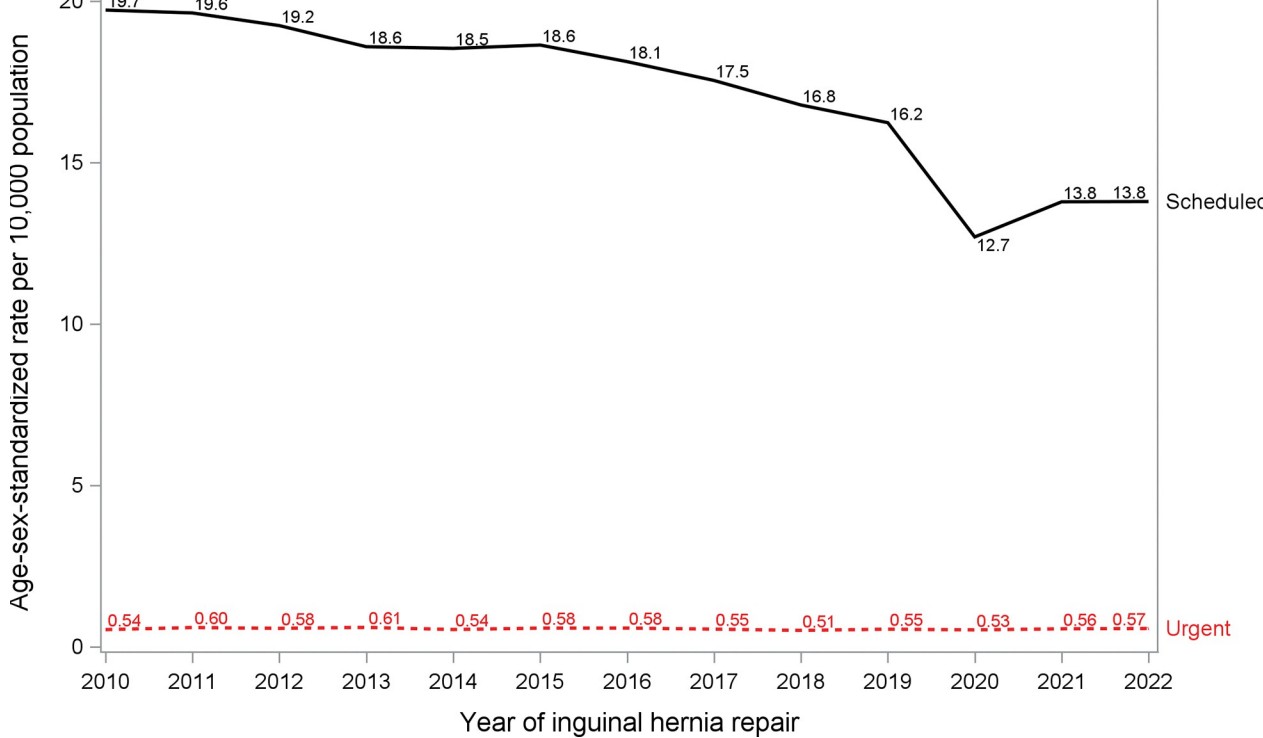

**Fig 1. Hernia repair by setting.** Number (A) and age-sex standardized rate (B) of scheduled (e.g. elective) and urgent inguinal hernia repairs over time. The 2019 Ontario census counts were used for standardization. Age groups were 18–29 years, 30–44 years, 45–59 years, and 60+ years for males and 18–44 years, 45–59 years, and 60+ years for females).

**Table 2. Number of scheduled hernia repairs over time relative to expectation.**

| | Wave as a predictor of the residual (e.g. the effect of wave on the average count of hernia repairs per week) | Percent change: (Observed–Predicted)/ Predicted * 100 | Total less-than-expected (negative) or more-than-expected (positive) number of hernia repairs. Average effect of wave on the number of repairs per week × duration of wave in weeks (320 weeks for pre-pandemic period; 27 weeks in wave 1; 26 weeks in wave 2; 22 weeks in wave 3; 19 weeks in wave 4; 13 weeks in wave 5; and 41 weeks in wave 6) |
|---|---|---|---|
| Wave | Coefficient (SE)[a] | % change[b] | Coefficient (SE) |
| 0 (intercept) | -0.03 (3.94) | 0.02 | -10.43 (1,259.72) |
| 1 | -174.91 (14.11)*** | -42.29 | -4,722.45 (381.04) |
| 2 | 18.57 (14.36) | 18.14 | 482.90 (373.38) |
| 3 | -92.49 (15.52)*** | -21.36 | -2,034.80 (341.47) |
| 4 | 0.21 (16.63) | 0.29 | 3.99 (315.94) |
| 5 | -144.57 (19.92)*** | -25.88 | -1,879.43 (259.01) |
| 6 | -0.29 (11.68) | 1.47 | -11.84 (478.93) |
| Total (waves 1–6) | - | - | **-8,162 (-9,912 to -6,411)** |

[a] Example interpretation: during wave 3, there was an average reduction in the number of inguinal hernia repairs of -92.5 (95% confidence interval -123, -62) per week

[b] Example interpretation: during wave 3, there was a 21.4% reduction in the number of inguinal hernia repairs per week

[c] Example interpretation: during wave 3, there were -92.49 procedures/week * 22 weeks = 2,034.8 fewer procedures than expected. Since the pandemic started (March 1, 2020 until December 31, 2022), there was a net 8,162 inguinal hernia repairs less-than-expected.

*p<0.05

**p<0.001

***p<0.0001

outpatient; 8.2km for non-urgent inpatient), but markedly higher for patients choosing treatment at Shouldice Hospital (median 59km). Characteristics for urgent admissions are reported in Table S2 in S1 File.

To examine whether patient characteristics changed over time, we restricted to patients receiving surgery in the outpatient setting or at Shouldice Hospital. Compared with the pre-pandemic period (March 2017-December 2019), patients receiving surgery in the COVID-19 era (March 2020-December 2022) were older, were less likely to be female [aOR 0.92 (0.87–0.97) for outpatients only], had higher ASA score [aOR 2.13 (1.93–2.35) for III vs I-II at Shouldice Hospital; aOR 1.24 (1.20–1.28) for III vs I-II for outpatients], were more likely to comprise patients with obesity [aOR 1.21 (1.07–1.38) for outpatients], were more likely to receive mesh [aOR 2.63 (1.46–4.39) at Shouldice Hospital; aOR 1.36 (1.28–1.44) for outpatient], and were less likely to comprise of patients residing in the highest material deprivation [aOR 0.87 (0.79–0.97) at Shouldice Hospital and aOR 0.90 (0.86–0.95) for outpatients for highest vs lowest deprivation] (Table S3 in S1 File). Patients treated at Shouldice Hospital in the COVID-19 era were more likely to have greater comorbidity [aOR 1.36 (1.08–1.70) for 2 vs none], but less likely for outpatients [aOR 0.80 (0.71–0.90) for 3+ vs none]. Travel distance did not increase substantively in the COVID-19 era. In the outpatient setting, use of laparoscopy for inguinal repair increased in the COVID-19 era, but this was likely driven by an increasing trend throughout the pre-pandemic period (Fig S3 in S1 File).

### Time until repair

The median time until treatment for scheduled inguinal repairs increased from 90 (IQR 48, 156) days in 2019 to 117 (54, 212) days in 2020 (Table S4 in S1 File). Despite a reduction to 105 (56, 190) days in 2021, the time until surgery increased further to 133 (69, 213) days by 2022.

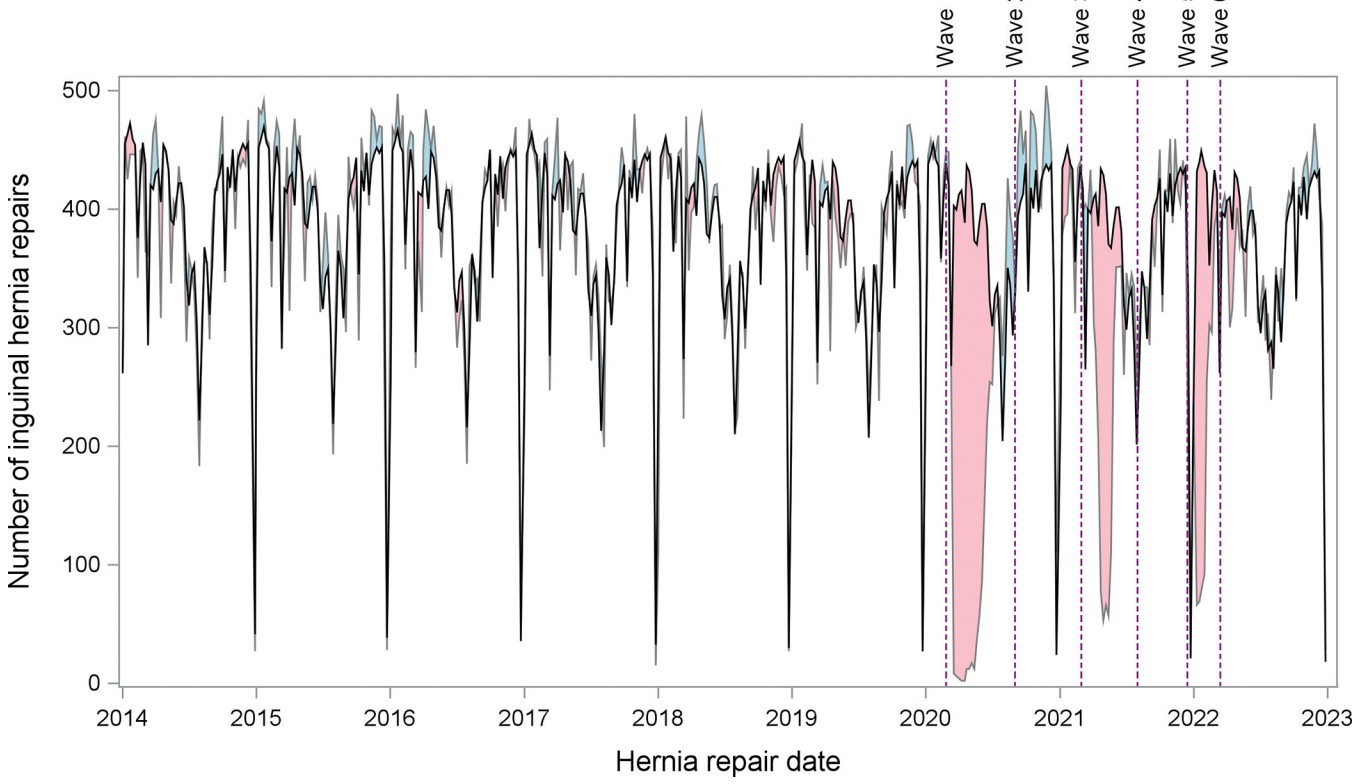

**Fig 2. Weekly scheduled inguinal hernia repair volumes over time.** Comparison of observed versus expected number of elective (scheduled) inguinal hernia repairs over time. The solid black line is the expected number of repairs per week. The pink shaded region below the line of expectation represents the total number of repairs in that was lower than expected (observed < expected). The blue shaded region above the line of expectation represents the total number of repairs that was greater than expected (observed > expected).

Compared with the pre-COVID-19 era, the COVID-19 era was associated with a longer time until surgery for patients receiving repairs at both Shouldice Hospital and outpatient hospital settings [OR 1.16 (1.14–1.17) and OR 1.12 (1.11–1.12), respectively] (Table S3 in S1 File).

## Hospital length of stay

Hospital length of stay was longest at the Shouldice Hospital [median 92 (92, 92) hours in 2019], followed by urgent inpatient procedures [median 52.5 (26.6, 97.4) hours in 2019] and scheduled inpatient procedures [median 28.2 (24.2, 37.6) days in 2019]. Compared with the 3 years before the pandemic, the length of stay at the Shouldice Hospital was reduced to 68 hours, which still exceeds average LOS for urgent repairs in the equivalent time period (Fig S4 in S1 File).

## Outcomes

90-day readmissions: A total 6471 (2.6%) scheduled inguinal hernia repairs were associated with a 90-day readmission (Table S4 in S1 File). After adjustment, 90-day readmission was less likely in the COVID-19 era [aOR 0.88 (0.81–0.96)] (Table 4). Other predictors of 90-day readmission include older age [aOR 1.36 (1.31–1.40)], higher ASA [aOR 1.72 (1.55–1.91) for ASA III and aOR 2.97 (2.58–3.42) for ASA IV-V versus I-II), repeat repair (aOR 1.24 (1.08–1.42)], higher comorbidity [aOR 2.36 (1.97–2.83) for 3+ versus none], and greater material deprivation [aOR 1.19 (1.05–1.36) for highest versus lowest deprivation].

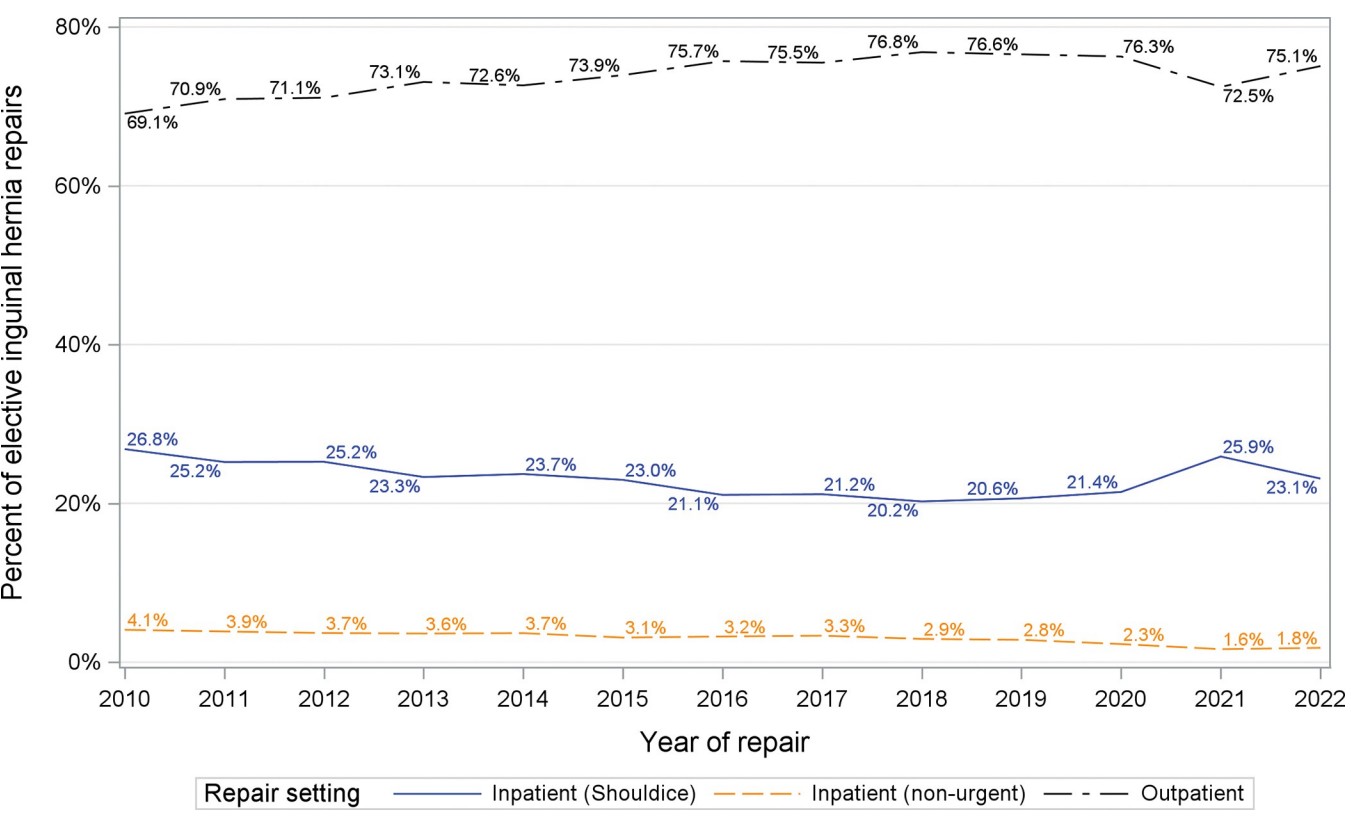

**Fig 3. Scheduled hernia repair by setting.** Percent of hernia repairs over time performed in the outpatient or inpatient setting.

90-day readmission was less likely in the COVID-19 era for patients receiving surgery in the outpatient setting [aOR 0.91 (0.83–0.99)] and at Shouldice Hospital [aOR 0.74 (0.58–0.95)], but there was no difference in the risk of 90-day readmission over time by setting [p = 0.09 for interaction between setting (Shouldice Hospital versus outpatient) and time period (COVID-19 era versus pre-COVID-19 era)].

1-year reoperations: A total of 4,116 (1.8%) primary scheduled inguinal repairs were associated with a reoperation within 1 year (Table S4 in S1 File). After adjustment, reoperations were less likely in the COVID-19 era [aOR 0.90 (0.80–1.02)] and for patients receiving surgery at Shouldice Hospital [aOR 0.71 (0.60–0.83)] (Table 4). Reoperations were more likely for older patients [aOR 1.05 (1.01–1.09)], females [aOR 1.30 (1.08–1.58)], rural residents [aOR 1.24 (1.05–1.46)], patients with higher ASA score [aOR 1.34 (1.05–1.70) for IV-V vs I-II], and higher comorbidity [aOR 1.78 (1.27–2.50) for 3+ vs none].

After adjustment, there was no difference in the change of 1-year reoperation between the COVID-19 era and the pre-COVID-19 era between patients treated at Shouldice Hospital versus the outpatient public setting (p = 0.17 for interaction).

## Discussion

In the present study, we observed pandemic-related reduction in scheduled inguinal hernia repair, longer wait-times for surgery, and changing patient demographics, but no evidence that patient outcomes worsened. Although there was some regional variability, the provision of outpatient care appears to have been maximized in most Ontario hospitals; when Shouldice Hospital is excluded, 96% of cases were performed in the outpatient setting. However, as

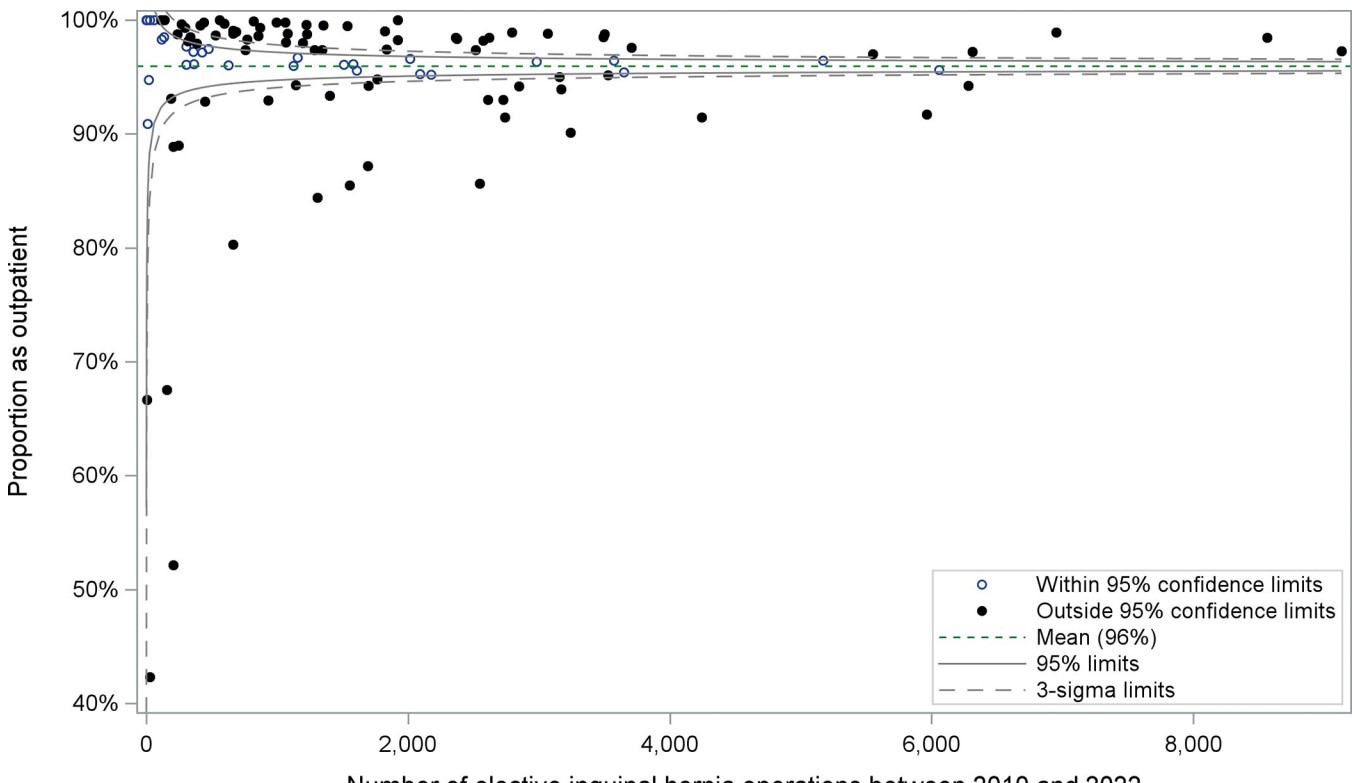

**Fig 4. Regional variability in the use of outpatient hernia repair for scheduled inguinal hernia repair (excluding Shouldice Hospital).** Funnel plot showing proportion of scheduled inguinal hernia repairs by hospital performed in the outpatient setting in relation to hernia repair volumes during the study period.

Shouldice Hospital performs a quarter of all hernia repairs in the province—and delivers these exclusively as inpatient cases—there is a sizable impact of the facility on Ontario's overall provincial outpatient rate. We observed that the increasing trend in outpatient surgery only declined slightly during the pandemic, while the decline in public inpatient surgery continued. The declining trend in private inpatient surgery (Shouldice Hospital) was reversed during the pandemic, and the length of stay was shortened despite treating more patients with comorbidity.

When examining measures of access and outcomes, we found that the pandemic was associated with longer average time until surgery, but did not increase rates of 90-day readmissions, or 1-year revisions. We observed no impacts on the rate of urgent repairs, and found that longer time until repair was not associated with worse outcomes. Despite this, the 8,162 fewer-than-expected scheduled inguinal hernia repairs performed during the pandemic period is concerning. Volume reductions coincided with surgical slowdowns in response to Ontario's COVID-19 Waves 1, 3 and 5 but surgical volumes subsequently returned to pre-pandemic levels. Studies from the US, Sweden, and Scotland demonstrated similar reductions in scheduled hernia repair during the early pandemic but also a transitory reduction in urgent procedures [11–13]. Changes in patient selection and preferred surgical approach (favouring open) was observed in one international survey of surgeons, yet we and others did not observe any overt changes in approach [13, 14].

Ontario's Shouldice Hospital plays a unique role in our study results, as an 89-bed private for-profit hospital specializing in hernia repair [6]. Surgical services at Shouldice Hospital are publicly funded, as legislated by the Canada Health Act, but patients pay out-of-pocket for the

**Table 3. Patient demographics of scheduled inguinal hernia repair by setting.**

| | Inpatient (Shouldice Hospital) | Outpatient (public hospital) | Inpatient (public hospital)[a] |
|---|---|---|---|
| **Total (all non-urgent)** | **57,755 (22%)** | **183,990 (71%)** | **7,797 (3%)** |
| Age (years), mean | 57.1 (SD 14.5) | 59.9 (SD 15.6) | 70.6 (SD 14.2) |
| Sex | | | |
| Male | 55197 (96%) | 169329 (92%) | 7030 (90%) |
| Female | 2576 (4.5%) | 14661 (8.0%) | 767 (9.8%) |
| American Society of Anesthesiologists score | | | |
| I-II | 54218 (94%) | 111790 (61%) | 1892 (24%) |
| III | 3476 (6%) | 62535 (34%) | 3866 (50%) |
| IV | 61 9<1% | 9637 (5.2%) | 2030 (26%) |
| V | 0 (0%) | 28 (<1%) | 9 (<1%) |
| Obesity | | | |
| No | 57753 (100%) | 181970 (99%) | 7592 (97%) |
| Yes | <6 | 2020 (1.1%) | 205 (2.6%) |
| Comorbidity score | | | |
| 0 | 50100 (87%) | 152870 (83%) | 4279 (55%) |
| 1 | 6625 (11%) | 23032 (13%) | 1848 (24%) |
| 2 | 821 (1.4%) | 5089 (2.8%) | 838 (11%) |
| 3+ | 209 (0.4%) | 2999 (1.6%) | 832 (11%) |
| Travel distance (km), median IQR | 58.9 (18.4, 136) | 9.3 (3.9, 24.9) | 8.2 (3.4, 22.1) |
| Time since diagnosis (months) | 2.5 (1.5, 4.4) | 2.8 (1.4, 5.3) | 3.1 (1.4, 6.1) |
| Hospital length of stay (hours) | 92.0 (68.0, 92.0) | n/a | 28.2 (24.9, 35.7) |
| Type of repair | | | |
| Laparoscopic | 0 (0%) | 35789 (19%) | 1114 (14%) |
| Open (other technique) | 422 (1%) | 145745 (79%) | 6551 (84%) |
| Open (special technique) | 57333 (99%) | 2456 (1%) | 132 (2%) |
| Use of mesh | | | |
| No mesh | 57516 (99%) | 14704 (8%) | 896 (11%) |
| Mesh | 239 (0.4%) | 169286 (92%) | 6901 (89%) |
| Recurrent repair | 2997 (5.2%) | 951 (12%) | 861 (11%) |
| Rurality | | | |
| Urban | 51389 (90%) | 154624 (85%) | 6762 (87%) |
| Rural | 6022 (10%) | 28160 (15%) | 996 (13%) |
| Deprivation | | | |
| 1 (least marginalized) | 17520 (31%) | 40305 (22%) | 1645 (21%) |
| 2 | 13770 (24%) | 37900 (21%) | 1551 (20%) |
| 3 | 10922 (19%) | 35658 (20%) | 1435 (19%) |
| 4 | 8667 (15%) | 34530 (19%) | 1577 (21%) |
| 5 (most marginalized) | 6030 (11%) | 32271 (18%) | 1477 (19%) |
| Ethnic diversity | | | |
| 1 (least diverse) | 10237 (18%) | 42819 (24%) | 1702 (22%) |
| 2 | 10580 (19%) | 37830 (21%) | 1531 (20%) |
| 3 | 12424 (22%) | 33974 (19%) | 1495 (19%) |
| 4 | 13088 (23%) | 32265 (18%) | 1520 (20%) |
| 5 (most diverse) | 10580 (19%) | 33776 (19%) | 1437 (19%) |
| Weekday | | | |
| Sunday | 12642 (22%) | 191 (<1%) | 27 (<1%) |

(*Continued*)

**Table 3.** (Continued)

| | Inpatient (Shouldice Hospital) | Outpatient (public hospital) | Inpatient (public hospital)[a] |
|---|---|---|---|
| Monday | 12417 (22%) | 32425 (18%) | 1358 (17%) |
| Tuesday | 10345 (18%) | 45446 (25%) | 2112 (27%) |
| Wednesday | 10235 (18%) | 36958 (20%) | 1305 (17%) |
| Thursday | 9996 (17%) | 38114 (21%) | 1694 (22%) |
| Friday | 2104 (4%) | 30625 (17%) | 1286 (16%) |
| Saturday | 16 (<1%) | 231 (<1%) | 15 (<1%) |
| Year of repair (row percents) | | | |
| 2010 | 5291 (26%) | 13620 (67%) | 804 (4%) |
| 2011 | 5054 (25%) | 14202 (69%) | 775 (4%) |
| 2012 | 5062 (25%) | 14252 (69%) | 737 (4%) |
| 2013 | 4603 (23%) | 14426 (71%) | 713 (4%) |
| 2014 | 4749 (23%) | 14560 (71%) | 733 (4%) |
| 2015 | 4706 (22%) | 15153 (72%) | 636 (3%) |
| 2016 | 4282 (20%) | 15382 (73%) | 657 (3%) |
| 2017 | 4250 (21%) | 15165 (73%) | 671 (3%) |
| 2018 | 3975 (20%) | 15085 (75%) | 575 (3%) |
| 2019 | 4003 (20%) | 14853 (74%) | 547 (3%) |
| 2020 | 3316 (21%) | 11801 (73%) | 353 (2%) |
| 2021 | 4421 (25%) | 12366 (70%) | 280 (2%) |
| 2022 | 4043 (22%) | 13125 (72%) | 316 (2%) |

[a] characteristics of urgent inpatient repairs (all public hospital) are reported in Table S2 in S1 File.

Column percents are shown, except where indicated

subsequent inpatient stay required. Average LOS for hernia repairs at Shouldice Hospital were 92 hours, longer than that for even urgent patients (52 hours). Although studies have demonstrated superior outcomes following hernia repair at Shouldice Hospital compared with public hospitals, this may be explained by patient selection and potentially greater surgeon experience rather than a protracted length of stay [15, 16]. The Shouldice Hospital moderately reduced their LOS during the pandemic period to either accommodate more surgeries or as a control measure against SARS-CoV-2 infection, but their outcomes were unaffected despite treating patients with more comorbidities and more advanced age than pre-pandemic.

## Limitations

One limitation of the present study is the surveilled population was excluded. Wait-list information for patients who have not received repair are unavailable and we are unaware of any validation studies that examine the accuracy of hernia diagnoses that have not been repaired (e.g. the watch-and-wait approach) [17]. This may also contribute to a longer time from diagnosis until surgery, signals of which were observed as early as 2017. Whether this is due to increased use of non-surgical management remains unknown. A second limitation is the absence of a true date of diagnosis. Our proxy definition had reasonable face validity compared with anecdotal expectations of wait-times, including wait-times of near-zero for urgent repairs. Third, it is possible that the incidence of symptomatic hernias decreased during the COVID-19 pandemic due to reductions on physical activity and work, but this is unlikely to account for a meaningful portion of the reduction in hernia repairs observed. Fourth is the lack of data

**Table 4. Outcomes following scheduled inguinal repair.**

| | 90-day readmission | 1-year reoperation |
|---|---|---|
| | OR (95% CI) | OR (95% CI) |
| **Crude** | N = 2609 events | N = 1301 events |
| COVID-19 era vs pre-COVID-19 era | 0.92 (0.85–0.99) | 0.90 (0.80–1.01) |
| **Adjusted[a]** | N = 2473 events | N = 1251 events |
| COVID-19 era vs pre-COVID-19 era | 0.88 (0.81–0.96) | 0.90 (0.80–1.02) |
| Age (per 10 years) | 1.36 (1.31–1.40) | 1.05 (1.01–1.10) |
| Sex (female vs. male) | 0.96 (0.82–1.12) | 1.30 (1.08–1.58) |
| Rurality (rural vs. urban) | 0.99 (0.86–1.13) | 1.24 (1.05–1.46) |
| Setting | | |
| Outpatient | 1.0 (ref) | 1.0 (ref) |
| Inpatient (Shouldice Hospital) | NR[b] | 0.71 (0.60–0.83) |
| Inpatient (public hospital) | 1.72 (1.47–2.01) | 0.84 (0.60–1.18) |
| Time until repair (per month) | 1.00 (0.98–1.01) | 0.99 (0.97–1.01) |
| ASA score | | |
| I-II | 1.0 (ref) | 1.0 (ref) |
| III | 1.72 (1.55–1.91) | 1.23 (1.08–1.41) |
| IV-V | 2.97 (2.58–3.42) | 1.34 (1.05–1.70) |
| Obesity vs no obesity | 1.25 (0.90–1.73) | 1.36 (0.86–2.14) |
| Repeat repair | 1.24 (1.08–1.42) | n/a |
| Charlson comorbidity score | | |
| None | 1.0 (ref) | 1.0 (ref) |
| 1 | 1.48 (1.33–1.64) | 1.23 (1.06–1.46) |
| 2 | 2.23 (1.91–2.60) | 1.57 (1.19–2.07) |
| 3+ | 2.36 (1.97–2.83) | 1.78 (1.27–2.50) |
| Material deprivation | | |
| 1 (least) | 1.0 (ref) | 1.0 (ref) |
| 2 | 1.00 (0.88–1.13) | 1.07 (0.90–1.27) |
| 3 | 1.04 (0.92–1.18) | 1.16 (0.98–1.38) |
| 4 | 1.13 (0.99–1.28) | 1.13 (0.94–1.35) |
| 5 (most) | 1.19 (1.05–1.36) | 1.16 (0.96–1.39) |
| Ethnic diversity | | |
| 1 (least) | 1.0 (ref) | 1.0 (ref) |
| 2 | 1.03 (0.91–1.16) | 1.08 (0.91–1.28) |
| 3 | 0.93 (0.81–1.06) | 1.00 (0.83–1.21) |
| 4 | 0.99 (0.87–1.14) | 0.91 (0.75–1.10) |
| 5 (most) | 0.92 (0.80–1.06) | 0.90 (0.74–1.09) |
| **Adjusted, by setting[a,c]** | | |
| COVID-19 era vs pre-COVID-19 era | | |
| Among outpatients | 0.91 (0.83–0.99) | 0.85 (0.75–0.97) |
| Among Shouldice Hospital | 0.74 (0.58–0.95) | 1.04 (0.77–1.40) |
| Interaction p-value | 0.09 | 0.17 |

Factors associated with 90-day readmission and1-year reoperation following scheduled inguinal hernia repair in the COVID-19 era (March 2020 through December 2022) compared with the pre-COVID-19 era (March 2017 through December 2019).

[a] odds ratios (OR) and 95% confidence intervals (CI) were adjusted for all variables shown

[b] not reported (NR) in the main table [OR 0.78 (0.68–0.90)] because of selection bias (readmissions for post-operative complications at Shouldice Hospital may not be captured by the Discharge Abstract Database).

[c] restricted only to patients receiving care in the outpatient public setting or at Shouldice Hospital

n/a–not applicable (restricted to primary repairs)

on patient quality of life and productivity, since patients with hernias are waiting longer for repair than before the pandemic. Fifth, 1-year reoperation as an outcome may be misclassified as a revision of an older repair since laterality was unavailable. Sixth, we may underestimate the rate of 90-day readmission for patients receiving care at Shouldice hospital because these readmissions would not be captured by the Discharge Abstract Database. Thus, we do not interpret differences in 90-day readmission observed between Shouldice Hospital and the public hospital setting. We expect our finding to generalize to jurisdictions with both public and private options for hernia repair.

## Conclusion

There was a sustained reduction in scheduled inguinal hernia repair since the start of the COVID-19 pandemic, but patient outcomes were unaffected. Surgical recovery and the subsequent redesign of surgical delivery must rely on the cost-effective benchmarks set by high quality outpatient care provided in public hospitals.

## Supporting information

**S1 File.**
(DOCX)

## Author Contributions

**Conceptualization:** Steven Habbous, David Gomez, David Urbach, Erik Hellsten.

**Data curation:** Steven Habbous.

**Formal analysis:** Steven Habbous.

**Investigation:** Steven Habbous, David Gomez, David Urbach, Erik Hellsten.

**Methodology:** Steven Habbous, David Gomez, David Urbach, Erik Hellsten.

**Project administration:** Steven Habbous.

**Supervision:** Erik Hellsten.

**Validation:** Steven Habbous, David Gomez, David Urbach, Erik Hellsten.

**Visualization:** Steven Habbous, David Urbach, Erik Hellsten.

**Writing – original draft:** Steven Habbous.

**Writing – review & editing:** David Gomez, David Urbach, Erik Hellsten.

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
