## [Decision Letter · Decision Letter 0]

18 Oct 2023

PONE-D-23-25270Scheduled and urgent inguinal, umbilical, and femoral hernia repair in Ontario, Canada between 2010 and 2022: population-based cross sectional analysis of trends and outcomesPLOS ONE

Dear Dr. Habbous,

Thank you for submitting your manuscript to PLOS ONE. After careful consideration, we feel that it has merit but does not fully meet PLOS ONE’s publication criteria as it currently stands. Therefore, we invite you to submit a revised version of the manuscript that addresses the points raised during the review process.

We look forward to receiving your revised manuscript.

Kind regards,

Franck Katembo Sikakulya

Academic Editor

PLOS ONE

Journal Requirements:

Did you know that depositing data in a repository is associated with up to a 25% citation advantage (https://doi.org/10.1371/journal.pone.0230416)? If you’ve not already done so, consider depositing your raw data in a repository to ensure your work is read, appreciated and cited by the largest possible audience. You’ll also earn an Accessible Data icon on your published paper if you deposit your data in any participating repository (https://plos.org/open-science/open-data/#accessible-data).

3. You indicated that ethical approval was not necessary for your study. We understand that the framework for ethical oversight requirements for studies of this type may differ depending on the setting and we would appreciate some further clarification regarding your research. Could you please provide further details on why your study is exempt from the need for approval and confirmation from your institutional review board or research ethics committee (e.g., in the form of a letter or email correspondence) that ethics review was not necessary for this study? Please include a copy of the correspondence as an ""Other"" file.

**Additional Editor Comments:**

This paper contains important information, and therefore we strongly suggest they re-draft this paper along the lines suggested by reviewers

Reviewers' comments:

Reviewer's Responses to Questions

**Comments to the Author**

1. Is the manuscript technically sound, and do the data support the conclusions?

Reviewer #1: Partly

Reviewer #2: Yes

2. Has the statistical analysis been performed appropriately and rigorously? 

Reviewer #1: I Don't Know

Reviewer #2: Yes

3. Have the authors made all data underlying the findings in their manuscript fully available?

Reviewer #1: Yes

Reviewer #2: No

4. Is the manuscript presented in an intelligible fashion and written in standard English?

Reviewer #1: Yes

Reviewer #2: Yes

5. Review Comments to the Author

Reviewer #1: This paper reports the change in rate of hernia operations during the COVID-19 pandemic. Its major finding was that there were fewer operations performed, except in a single private hospital that performed more surgery and reduced its length of stay. Although the Abstract gives the decline in surgeries performed as numbers it does not give the percentage drop, which it should.

This paper is information heavy, and really is a case of information overload and displays a dogged determination to ensure the reader cannot see the wood for the trees. As the author’s concede in their Discussion, uniquely the 89 bed Shouldice Hospital that only specializes in hernia surgery performs 18.9% of all hernia surgeries in Ontario. Moreover, nearly 90% of all hernia operations performed in public hospitals are outpatient procedures. Therefore, the features that make this paper unique and of interest to international readers is the impact, if any, the COVID-19 pandemic had on these two practice innovations. The authors should re-draft their paper to make the changes in 1. in-patient public hospital hernia surgery, 2. out-patient public hospital hernia surgery, and 3. in-patient private specialist hernia surgery clearer. It took me a long time to sort all this out, and reading and re-reading the data to realize that the outcomes for all these three were NOT the same.

Hernia repair is arguably the quintessential elective surgery, as for most patients it is often only an inconvenience as strangulation and other life-threatening complications are relatively uncommon. For simplicity of presentation, it might be reasonable to combine all three, or as the risk of complications may not be the same for all of them, confining most of the analysis to inguinal hernia. The point being that what the reader wants to know are:

1. The increasing trend in out-patient surgery only declined slightly during the pandemic.

2. The declining trend in private in-patient surgery was reversed, and the length of stay shortened despite patients with more co-morbidity being treated.

3. The declining trend in public in-patient surgery continued.

4. What happened to the need for urgent repair? These data need to be qualified by / related to the duration of symptoms, the availability of private in-patient surgery, public in-patient, and out-patient surgery. My hypothesis would be that those patients who had symptoms the longest, and lived furthest from any of these available services or who could least afford to avail of them would be most likely to require urgent surgery. A logistic regression model might also consider other variables.

As I consider this paper needs a complete re-draft, I will not go through it in too much detail. However, I note that the Methods section does not mention the clinical Settings, and Table 1 is far too detailed, and it is not clear what the percentages provided are percentages of. Furthermore, the Figures have no adequate legends.

Reviewer #2: Abstract

1. Unsure how the study is a “case-control” study. This term is not used in the methods section of the manuscript.

2. The methods section of the abstract could use more detail and there is room within the 300 word limit to implement this.

Introduction

3. Conveying the high proportion of hernia repairs in Ontario that performed at the Shouldice Hospital would help convey why this institution of is of particular interest. As some analyses are related specifically to operations performed at Shouldice it would also be helpful for the hospital to appear in the objective.

4. It is clear from the introduction why trends in the volume of surgical hernia repairs may have changed due to the COVID-19 pandemic but not clear why outcomes following surgery may have been impacted. I would suggest the authors provide some background into the rationale for looking for changes in outcomes.

Methods

5. There is no mention of Shouldice Hospital in the Methods section despite it having it a separate subheading in the Results section. Please describe how in the methods how Shouldice will be analyzed separately.

6. Could the authors add more detail on the regression model, including on how autocorrelation was considered?

6. PLOS authors have the option to publish the peer review history of their article (what does this mean?). If published, this will include your full peer review and any attached files.

Reviewer #1: **Yes: **John Kellett

Reviewer #2: No

---

## [Author Response · Author response to Decision Letter 0]

15 Nov 2023

EDITOR COMMENTS:

Comment: This paper contains important information, and therefore we strongly suggest they re-draft this paper along the lines suggested by reviewers

Response: Thank you for the positive note. We ultimately restricted the manuscript only to inguinal hernias as per Reviewer #1’s comments. We reorganized the results section to improve the flow of the manuscript, which we hope will help the readers.

REVIEWER COMMENTS:

Reviewer: 1

Comment #1: This paper reports the change in rate of hernia operations during the COVID-19 pandemic. Its major finding was that there were fewer operations performed, except in a single private hospital that performed more surgery and reduced its length of stay. Although the Abstract gives the decline in surgeries performed as numbers it does not give the percentage drop, which it should.

Response: Thank you for the suggestion, we added the percent reduction (15%) in the abstract

Comment #2: This paper is information heavy, and really is a case of information overload and displays a dogged determination to ensure the reader cannot see the wood for the trees. As the author’s concede in their Discussion, uniquely the 89 bed Shouldice Hospital that only specializes in hernia surgery performs 18.9% of all hernia surgeries in Ontario. Moreover, nearly 90% of all hernia operations performed in public hospitals are outpatient procedures. Therefore, the features that make this paper unique and of interest to international readers is the impact, if any, the COVID-19 pandemic had on these two practice innovations. The authors should re-draft their paper to make the changes in 1. in-patient public hospital hernia surgery, 2. out-patient public hospital hernia surgery, and 3. in-patient private specialist hernia surgery clearer. It took me a long time to sort all this out, and reading and re-reading the data to realize that the outcomes for all these three were NOT the same.

Response: Thank you for the comment and we agree that there is a lot of information presented in this manuscript. To improve the clarity of the work, we removed umbilical and femoral hernia repairs and instead focus solely on inguinal hernia. We reorganized the results as follows, examining how the COVID-19 pandemic affected:

1. Surgical volumes and population rates over time

2. Use of hospital setting (e.g. outpatient versus public-inpatient versus Shouldice versus urgent)

3. Patient characteristics by setting and over time

4. Timeliness of care (e.g. time from diagnosis until surgery)

5. The provision of care (hospital length of stay)

6. Patient outcomes (e.g. readmission and revision)

To organize the paper further, in addition to providing more meaningful subheadings, we also replaced Table 1 with Table S2 (patient demographics by setting) and explicitly list out the different settings a patient can receive care early in the methods (Covariates subsection). One of the objectives of this work was to assess the quality of care, so it was important to report on the number and rate of urgent hernia repairs because it was conceivable that reduced elective procedures would result in worsening conditions necessitating urgent repair. For completeness, we also report the patient characteristics for the urgent group for the interested reader, but we relegate this to the supplement to reduce clutter in the prose. We also moved the main outcomes table to the prose (now Table 3). It was previously left to the supplement, but now that it’s only restricted to inguinal repairs, the manuscript is less cluttered.

Comment #3: Hernia repair is arguably the quintessential elective surgery, as for most patients it is often only an inconvenience as strangulation and other life-threatening complications are relatively uncommon. For simplicity of presentation, it might be reasonable to combine all three, or as the risk of complications may not be the same for all of them, confining most of the analysis to inguinal hernia. The point being that what the reader wants to know are:

1. The increasing trend in out-patient surgery only declined slightly during the pandemic.

2. The declining trend in private in-patient surgery was reversed, and the length of stay shortened despite patients with more co-morbidity being treated.

3. The declining trend in public in-patient surgery continued.

4. What happened to the need for urgent repair? These data need to be qualified by / related to the duration of symptoms, the availability of private in-patient surgery, public in-patient, and out-patient surgery. My hypothesis would be that those patients who had symptoms the longest, and lived furthest from any of these available services or who could least afford to avail of them would be most likely to require urgent surgery. A logistic regression model might also consider other variables.

As I consider this paper needs a complete re-draft, I will not go through it in too much detail. However, I note that the Methods section does not mention the clinical Settings, and Table 1 is far too detailed, and it is not clear what the percentages provided are percentages of. Furthermore, the Figures have no adequate legends.

Response: We have indeed taken this comment to heart, and have restricted the analyses to inguinal repairs in the revision. The first paragraph of the discussion that summarizes the findings now includes these points:

“We observed that the increasing trend in outpatient surgery only declined slightly during the pandemic, while the decline in public inpatient surgery continued. The declining trend in private inpatient surgery (Shouldice Hospital) was reversed during the pandemic, and the length of stay was shortened despite treating more patients with comorbidity.”

Since we removed femoral and umbilical hernias, the former Table 1 was not considered helpful, and was replaced with what was formerly supplementary Table S2 (patient characteristics by setting; now Table 2). The urgent inpatient group was left to the supplement as mentioned above. We have added “travel distance” to the list of covariates and report these in Table 2. Interestingly, there was no meaningful difference in travel distance for patients treated at public hospitals (median 9.3km for outpatient; 8.2km for non-urgent inpatient) compared with Shouldice Hospital (median 59km). As the rate of urgent repairs remained steady over time and is largely out of the control of the healthcare system, we did not examine this subgroup further.

Note that the figure legends are embedded within the prose rather than associated with the figure as per PLoS guidelines. We also clarify the meaning of the percentages in Table 2 in the revision.

 

Reviewer: 2

Abstract

Comment #1: Unsure how the study is a “case-control” study. This term is not used in the methods section of the manuscript.

Response: Thank you for noticing this, we have removed “case-control” from the methods section of the abstract. 

Comment #2: The methods section of the abstract could use more detail and there is room within the 300 word limit to implement this.

Response: Great point, we have extended the abstract to include more methodological details (a brief primer on Shouldice Hospital; regression method; covariates used for adjustment). 

Introduction

Comment #3: Conveying the high proportion of hernia repairs in Ontario that performed at the Shouldice Hospital would help convey why this institution of is of particular interest. As some analyses are related specifically to operations performed at Shouldice it would also be helpful for the hospital to appear in the objective.

Response: This is a good point. We have added the following to the concluding statement of the introduction section: “We examine where patients receive surgery, contrasting those treated at public hospitals with those treated at the Shouldice Hospital”. We also add that the Shouldice Hospital performs a large percentage of all inguinal repairs in the province, highlighting its importance. 

Comment #4: It is clear from the introduction why trends in the volume of surgical hernia repairs may have changed due to the COVID-19 pandemic but not clear why outcomes following surgery may have been impacted. I would suggest the authors provide some background into the rationale for looking for changes in outcomes.

Response: This is another good point. We have added the following to the concluding statement of the introduction section: “We assessed the change in patient characteristics and outcomes over time to ensure that the quality of care was not adversely affected due to hospital pressures imposed by the COVID-19 pandemic.”

Methods

Comment #5: There is no mention of Shouldice Hospital in the Methods section despite it having it a separate subheading in the Results section. Please describe how in the methods how Shouldice will be analyzed separately.

Response: Thank you for noting this. The “Shouldice” subheading in the results is no longer present. We added the following to the methods section to clearly describe the breakdown of the cohort by setting:

“The repair setting was classified as outpatient if captured by the CIHI-NACRS database and inpatient if captured by CIHI-DAD. Inpatient procedures were further subclassified as Shouldice Hospital if the procedure was performed there; non-urgent inpatient; and urgent inpatient. Urgent operations were defined as an admission with entry code = E (admitted through the emergency department), admission category = U (urgent), or if the patient arrived by ambulance.”

Comment #6: Could the authors add more detail on the regression model, including on how autocorrelation was considered?

Response: Certainly. The regression model used the weekly count as a predictor. The number of procedures performed in a given week is not expected to depend on the number of procedures performed in the previous week, thus, we did not consider autocorrelation. Instead, we expected the week number itself (1-52) to more aptly predict week-to-week variability, because this is better suited to incorporate consistent trends related to holidays and vacation times. Moreover, autocorrelation models require assumptions about the nature of the correlation. The model we used had excellent fit (very small and acceptable error), so we did not opt to search for a more complex model. Lastly, the estimated number of hernia repairs “missed” during the pandemic was calculated quite accurately using the regression model.

---

## [Decision Letter · Decision Letter 1]

27 Nov 2023

PONE-D-23-25270R1Scheduled and urgent inguinal hernia repair in Ontario, Canada between 2010 and 2022: population-based cross sectional analysis of trends and outcomesPLOS ONE

Dear Dr. Habbous,

Thank you for submitting your manuscript to PLOS ONE. After careful consideration, we feel that it has merit but does not fully meet PLOS ONE’s publication criteria as it currently stands. Therefore, we invite you to submit a revised version of the manuscript that addresses the points raised during the review process.

We look forward to receiving your revised manuscript.

Kind regards,

Franck Katembo Sikakulya

Academic Editor

PLOS ONE

Journal Requirements:

Additional Editor Comments:

Please find attached the comments from reviewers for you t improve the draft before its consideration.

Reviewers' comments:

Reviewer's Responses to Questions

**Comments to the Author**

1. If the authors have adequately addressed your comments raised in a previous round of review and you feel that this manuscript is now acceptable for publication, you may indicate that here to bypass the “Comments to the Author” section, enter your conflict of interest statement in the “Confidential to Editor” section, and submit your "Accept" recommendation.

Reviewer #1: All comments have been addressed

Reviewer #2: All comments have been addressed

2. Is the manuscript technically sound, and do the data support the conclusions?

Reviewer #1: Yes

Reviewer #2: (No Response)

3. Has the statistical analysis been performed appropriately and rigorously? 

Reviewer #1: Yes

Reviewer #2: Yes

4. Have the authors made all data underlying the findings in their manuscript fully available?

Reviewer #1: Yes

Reviewer #2: No

5. Is the manuscript presented in an intelligible fashion and written in standard English?

Reviewer #1: Yes

Reviewer #2: Yes

6. Review Comments to the Author

Reviewer #1: This paper is much improved, and I now have only a few minor issues:

1. In the Abstract please put in 15% afte '... 8,162 {15%) fewer ....'

2. Although the setting is mentioned in the Results, I think it should also have a separate heading in the Methods - e.g. Setting - patients management in three clinical setting was examined: Public hospital in-patient care, Private hospital in-patient care, and Public hospital out-patient care

3. The Results are presented rather unusually. Most papers start the Results with patient demographics usually presented in Table, followed by the rest of the Results. This paper does not do that, but starts with Table 2, and then introduces the demographics etc later on, with Table 1 following Table 2. This is most confusing. I realise there is a lot in Table 1. Therefore it might be better to split the demographics off as a separate table. However, there is no reason not to refer to Table 1, then Table 2, and then go back to Table 1 is needed.

Reviewer #2: I would like to thank the authors for their efforts to revise the manuscript per reviewer's comments.

7. PLOS authors have the option to publish the peer review history of their article (what does this mean?). If published, this will include your full peer review and any attached files.

Reviewer #1: **Yes: **John Kellett

Reviewer #2: No

---

## [Author Response · Author response to Decision Letter 1]

30 Nov 2023

REVIEWER #1: 

This paper is much improved, and I now have only a few minor issues:

Comment #1: In the Abstract please put in 15% after '... 8,162 {15%) fewer ....'

Response: Thank you for the suggestion, we moved the percent to after the word “fewer”.

Comment #2: Although the setting is mentioned in the Results, I think it should also have a separate heading in the Methods - e.g. Setting - patients management in three clinical setting was examined: Public hospital in-patient care, Private hospital in-patient care, and Public hospital out-patient care

Response: This is a good suggestion to improve the clarity of the methods. In the abstract, we added the following sentence: “Patients managed in three clinical settings were examined: public hospital in-patient, private hospital in-patient (Shouldice Hospital), and public hospital out-patient.” In the manuscript, we split out the “Covariates” subsection of the methods into “Setting” [describes patient groupings into public inpatient, private inpatient, public outpatient] and “Covariates [describes the other covariates]. 

Comment #3: The Results are presented rather unusually. Most papers start the Results with patient demographics usually presented in Table, followed by the rest of the Results. This paper does not do that, but starts with Table 2, and then introduces the demographics etc later on, with Table 1 following Table 2. This is most confusing. I realise there is a lot in Table 1. Therefore it might be better to split the demographics off as a separate table. However, there is no reason not to refer to Table 1, then Table 2, and then go back to Table 1 is needed.

Response: Thank you for noticing this. Indeed we previously had Table 2 listed as Table 1 (for the reason you describe), but ended up moving it to align with the presentation of the results. We added a new Table 1 (“Cohort characteristics”), which is a simplified (e.g. aggregated) descriptive statistics table the most pertinent cohort factors.

REVIEWER #2:

Comment #3: I would like to thank the authors for their efforts to revise the manuscript per reviewer's comments.

Response: Thank you your help improving this manuscript.

---

## [Decision Letter · Decision Letter 2]

4 Dec 2023

PONE-D-23-25270R2Scheduled and urgent inguinal hernia repair in Ontario, Canada between 2010 and 2022: population-based cross sectional analysis of trends and outcomesPLOS ONE

Dear Dr. Habbous,

Thank you for submitting your manuscript to PLOS ONE. After careful consideration, we feel that it has merit but does not fully meet PLOS ONE’s publication criteria as it currently stands. Therefore, we invite you to submit a revised version of the manuscript that addresses the points raised during the review process.

We look forward to receiving your revised manuscript.

Kind regards,

Franck Katembo Sikakulya

Academic Editor

PLOS ONE

Journal Requirements:

Additional Editor Comments:

Please

Make changes as suggested by the reviewer

Thank you

Reviewers' comments:

Reviewer's Responses to Questions

**Comments to the Author**

1. If the authors have adequately addressed your comments raised in a previous round of review and you feel that this manuscript is now acceptable for publication, you may indicate that here to bypass the “Comments to the Author” section, enter your conflict of interest statement in the “Confidential to Editor” section, and submit your "Accept" recommendation.

Reviewer #1: All comments have been addressed

2. Is the manuscript technically sound, and do the data support the conclusions?

Reviewer #1: Yes

3. Has the statistical analysis been performed appropriately and rigorously? 

Reviewer #1: Yes

4. Have the authors made all data underlying the findings in their manuscript fully available?

Reviewer #1: Yes

5. Is the manuscript presented in an intelligible fashion and written in standard English?

Reviewer #1: Yes

6. Review Comments to the Author

Reviewer #1: Thank you for your revisions. However, there now appear to be TWO Table 3's! Please correct this, and ensure that all Tables in the text or correct

7. PLOS authors have the option to publish the peer review history of their article (what does this mean?). If published, this will include your full peer review and any attached files.

Reviewer #1: **Yes: **John Kellett

---

## [Author Response · Author response to Decision Letter 2]

6 Dec 2023

Additional Editor Comments:

Comment #1: Please make changes as suggested by the reviewer. Thank you

Response: We have made these changes (checked numbering of all tables and figures and supplement). 

REVIEWER #1: 

This paper is much improved, and I now have only a few minor issues:

Comment #1: Thank you for your revisions. However, there now appear to be TWO Table 3's! Please correct this, and ensure that all Tables in the text or correct

Response: Thank you for catching this! We renamed the second Table 3 to Table 4, corrected this in the title page meta-data, and checked all references in the main body to any figures or tables, including the supplement to ensure we haven’t missed any others. 

Thank you again for your diligence in reviewing this manuscript.

---

## [Decision Letter · Decision Letter 3]

10 Dec 2023

Scheduled and urgent inguinal hernia repair in Ontario, Canada between 2010 and 2022: population-based cross sectional analysis of trends and outcomes

PONE-D-23-25270R3

Dear Dr. Steven Habbous,

We’re pleased to inform you that your manuscript has been judged scientifically suitable for publication and will be formally accepted for publication once it meets all outstanding technical requirements.

Kind regards,

Franck Katembo Sikakulya

Academic Editor

PLOS ONE

---

## [Editor Report · Acceptance letter]

14 Dec 2023

PONE-D-23-25270R3 

PLOS ONE

Dear Dr. Habbous, 

I'm pleased to inform you that your manuscript has been deemed suitable for publication in PLOS ONE. Congratulations! Your manuscript is now being handed over to our production team.

Kind regards, 

on behalf of

Dr Franck Katembo Sikakulya 

Academic Editor

PLOS ONE